# Diversify the syllabi: Underrepresentation of female authors in college course readings

**Jenine K. Harris** *, Merriah A. Croston, Ellen T. Hutti, Amy A. Eyler

Brown School, Washington University in St. Louis, St. Louis, Missouri, United States of America

* harrisj@wustl.edu

## Abstract

Emerging evidence demonstrates that female-authored publications are not well represented in course readings in some fields, resulting in a syllabi gender gap. Lack of representation may decrease student awareness of opportunities in professional fields and disadvantage the career success of female academics. We contribute to the evidence on the syllabi gender gap by: 1) quantifying the extent to which female authors are represented in assigned course readings; 2) examining representation of female authors by gender of instructor and discipline; and 3) comparing female representation in syllabi with the workforce and with representation as authors of peer-reviewed journal articles. From a list of courses offered in 2018–2019 at Washington University in St. Louis, we selected a stratified random sample of course syllabi from four disciplines (humanities; social science; science, technology, engineering, and mathematics; and other). We coded the gender of course instructors and course reading authors using the genderize application programming interface. We examined representation of female authors at the reading, course, and discipline level using descriptive statistics and data visualization. The final sample included 2435 readings from 129 unique courses. The mean percentage of female authors per reading was 34.1%; 822 (33.8%) of readings were female-led (i.e., a female first or sole author). Female authorship varied by discipline, with the highest percentage of female-led readings in social science (40%). Female instructors assigned a higher percentage of readings with female first authors and readings with higher percentages of females on authorship teams. The representation of female authors on syllabi was lower than representation of females as authors in the peer-reviewed literature or in workforce. Adding to evidence of the syllabi gender gap, we found that female authors were underrepresented as sole and first authors and as members of authorship teams. Since assigned readings promote academic scholarship and influence workforce diversity, we recommend several strategies to diversify the syllabi through increasing awareness of the gap and improving access to female-authored publications.

## Introduction

Academic careers are built on producing scholarly work, promoting it, and having your work cited by others. However, there is a gender disparity in these mechanisms of career success.

---

**Data Availability Statement:** All relevant data are within the manuscript, Supporting Information files, and on GitHub: https://github.com/jenineharris/syllabusStudy.

**Funding:** The author(s) received no specific funding for this work.

**Competing interests:** The authors have declared that no competing interests exist.

Research in political science found little difference in the overall amount of scholarly productivity (e.g., submitted books, grants, articles) by academic men and women [1], however, studies across numerous disciplines find women less likely than men to submit and publish published peer-reviewed papers [1–3]. A portion of this difference can be accounted for by women submitting more book chapters compared to men [1, 4] but having lower rates of publication for those chapters [1]. Further, the system that supports academic journal publishing may perpetuate this gender gap as most peer-reviewers [5, 6] and editors of academic journals are men [7–9].

Even if women overcome the extant challenges to publishing, journal articles authored by women are less likely to be cited or used by others compared to articles by their male colleagues. In an analysis of 15 years of publications, Knobloch-Westerwick et al. found articles by female communication scientists received fewer citations than articles authored by males [10]. Other studies support these early findings, including studies of citation parity in the fields of ecology and evolution [11], psychology [12], political science [13], and library and information science [14]. In addition to having work cited by others, there are disparities in self-promotion. Men are more likely than women to cite their own work [15] with men self-citing 70% more than women in papers published between 1991 and 2011. Fewer citations translates to lower visibility of women's scholarship compared to men. These patterns of in scholarly productivity and visibility result in differences in salary (https://www.chronicle.com/article/Divvying-Up-the-Raise-Pool/45750/), promotion, and tenure in academia [3], contributing to the gender stratification [16] of academic career success [3, 4, 17, 18].

Students are the next generation of the workforce, including the academic workforce. Some students are entering fields where they are professionally underrepresented (e.g., female students in engineering). There is evidence that students are more likely to succeed if they are exposed to same-gender experts and peers [19, 20] and if they have role models and mentors with whom they share race and gender identities [21]. College students encounter experts, such as course instructors, in many ways. However, not every student will have identity-matched instructors. Exposure to experts with whom they identify can happen through other sources like course readings on syllabi. In addition to encouraging undergraduate students to adopt or retain a major, course readings on syllabi in doctoral courses may be socializing the next generation of faculty [17].

Unfortunately, early research suggests that the gender gap in publishing and citation also persists in assigned course readings [17, 22–25]. This "syllabi gender gap" may be due to several factors. First, more male authors are represented in course readings. In an analysis of syllabi in the field of international relations, more than 70% of assigned readings had all-male authorship teams, independent of instructor gender [26]. Also, female instructors may be less likely than male instructors to assign their own publications as course materials. In the same study, male faculty assigned twice as many self-authored readings compared to female faculty [26]. Some faculty may assign the most popular or classic "elite" papers in their field, which tend to have a large gender gap [27].

Addressing systemic issues contributing to gender stratification in academic scholarship and workforce success are complex, but initiatives such as improving gender diversity in course syllabi is a feasible step toward change. We aim to enhance the limited existing evidence on the syllabi gender gap by: 1) quantifying the extent to which female authors are represented in assigned course readings among courses offered at a large private university in the United States; 2) making comparisons of female author representation on syllabi by course instructor gender and course discipline; and 3) comparing female representation in syllabi to female representation in the academic literature and the workforce.

Information gained from this study contributes to the evidence of a syllabi gender gap, informs future studies, and can be used to substantiate recommendations for instructors and institutions to achieve equal gender representation within course syllabi.

## Methods

Sex and gender are complex and evolving concepts incorporating biology, identity, and culture. Definitions and measurement of sex and gender vary and are not used in a consistent way in society or in the scientific literature [28]. After carefully considering the terminology used in existing syllabi gender gap literature and the limitations and terminology from the software tool we used for the majority of coding, and after consulting with a gender scholar, the research team elected to use the term "gender" rather than "sex" to describe what we measured and, within gender, to include 3 categories: male, female, and non-binary.

### Data collection

To examine the gender of authors of college course readings, we collected syllabi of a subset of courses taught at Washington University in St. Louis (WUSTL) during the fall and spring of the 2018–2019 academic year. WUSTL is a large private secular university situated in St. Louis, Missouri, United States (US). The 2020 total enrollment was 13,654, which included 2,717 international students. The 2018–2019 WUSTL undergraduate student body was 53% female and 61% US White (https://diversity.wustl.edu/framework/) with 15% of students qualifying for the Federal Pell Grant Program that supports low-income students. In 2020, WUSTL was ranked the 19th best overall of national universities in the US, 31st in best global universities (https://www.usnews.com/best-colleges), and 61st best in undergraduate education in the US. WUSTL is highly selective, with 15% of applicants accepted in 2018 and 88% on-time graduation. The student body has more ethnic diversity than most national universities in the US (https://www.usnews.com/best-colleges/rankings/national-universities/campus-ethnic-diversity) but has a very little economic diversity (https://www.usnews.com/best-colleges/rankings/national-universities/economic-diversity) compared to other US universities. The national and global reputation of WUSTL suggest a high-quality institution that is a model for other Universities. The majority of WUSTL programs are housed on one of two main campuses, the Danforth campus and the Medical campus.

Our study was deemed *not human subjects research* by the Institutional Review Board (IRB ID: 202001078). Courses were identified using an administrative dataset that contained information for 4856 courses offered on the Danforth Campus in fall 2018 and spring 2019. From this dataset, we identified all unique courses (Fig 1). We took a random sample of 10% of courses stratified by the four disciplines [humanities; social science; science, technology, engineering, and mathematics (STEM); and other (business, law, design and visual arts)]. The sample included 71 from humanities, 60 from other disciplines, 58 from social science, and 51 from STEM. We obtained 80 of the 240 syllabi from a central syllabus repository maintained by the university. For the remaining 160 syllabi, we individually emailed instructors to introduce the study and request their syllabus. We received 124 emails in response, including out-of-office replies, statements that there was not a syllabus, 15 positive notes encouraging the research, one note stating "This is an improper request and I decline to respond. The gender of an author is of no significance and should not be considered," several syllabi from a semester or course not in the sample, and 68 syllabi from the sample. In all, we retrieved syllabi for 61.7% (n = 148) of the sampled courses. We excluded 19 syllabi from the analytic data set for a final sample size of 129 courses; we excluded a syllabus if it did not contain readings or if *all* readings had authors with unknown names or genders.

We examined the number of courses per discipline in the final sample and compared it to the sampling frame to determine whether there were any substantial differences in the percentage of syllabi we were able to obtain by discipline. We received 38 humanities syllabi out of 71 (53.5%), 20 other discipline syllabi out of 60 (33.3%), 44 social science syllabi out of 58

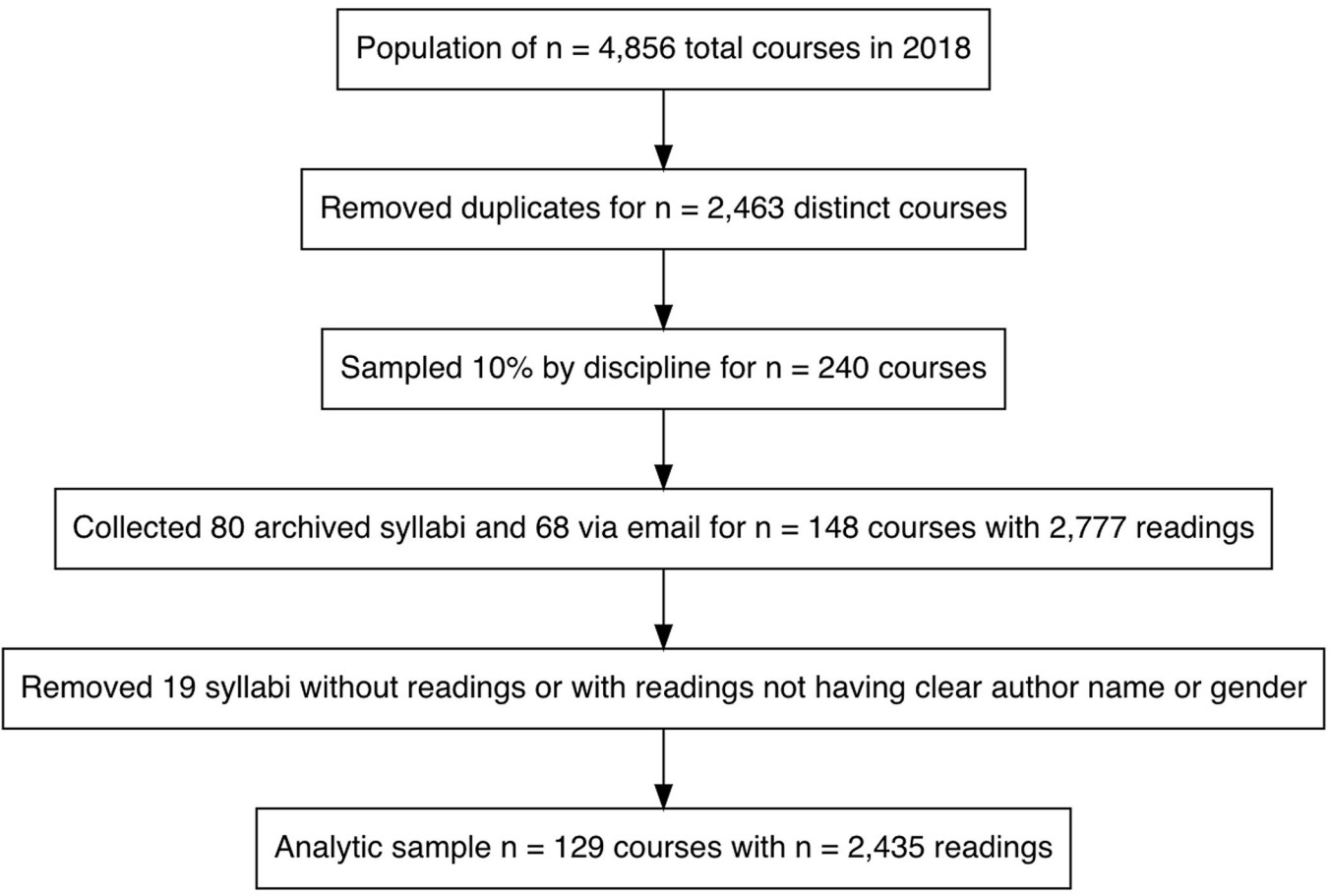

**Fig 1. Sampling flow chart to select 129 courses with 2435 readings from a college campus in 2018–2019.**

(75.9%), and 27 STEM syllabi out of 51 (52.9%). To determine whether the lower rate for other disciplines was due to excluding obtained syllabi, we examined the 19 syllabi removed from the initial sample. We found that 2 humanities courses, 4 other discipline courses, 4 social sciences courses, and 9 STEM courses were removed from the analytic sample.

Three research team members reviewed a portion of the syllabi to collect a list of readings. For each reading, the first name of each author was abstracted. If the first name of an author was not listed on a syllabus, the team member reviewing the syllabus conducted a Google search to find the first name. We randomly selected 15 syllabi to be double coded by one of the team members in order to assess reliability of identification of readings and author names. We found 95.8% agreement between the original coding and the second coding on which readings were contained in the syllabi and 97.8% agreement on the author first names. Data collection resulted in 2,435 readings (Fig 1).

## Coding author and instructor gender

After abstracting readings and authorship information from syllabi, we coded the gender of all course instructors and authors. Author gender and instructor gender were coded using the genderize application programming interface (API), which assigns a probability of a first name belonging to a male or female based on over 114 million entries of names from 242 countries around the world (https://genderize.io/our-data). Of note, the genderize application is unable

to distinguish nonbinary authors. Following Fox and Paine (2019) [11] we used the male or female gender classification by the genderize application if the classification had a probability of 95% or higher of being correct based on the genderize data. Consistent with recent studies of author gender, for those names with a probability of correct classification lower than 95%, two members of the research team looked up each author using Google and classified the author gender based on a pronoun, photo, or both [29, 30]. Each unsure author gender was classified as male, female, non-binary, or unknown. To ensure that this classification was reliable, a third research team member reclassified 10% of the unsure names. Of the 62 names reclassified, 3 (4.8%) did not match the initial classification.

## Other data sources

To compare gender representation in our sample of syllabi with gender representation in the workforce, we examined the distribution of gender among authors of papers indexed in JSTOR between 1545–2011, college graduates in the workforce, employed faculty, and those who recently earned doctorates. The JSTOR data were obtained from Table 1 in West et al. [2]. The college graduates data were obtained from the National Science Foundation (NSF) 2017 National Survey of College Graduates (https://ncsesdata.nsf.gov/datadownload/), which is a survey of people with at least a bachelor's degree. The survey is conducted every two to three years by the NSF and asks participants about education, employment, and demographic information. Gender distribution among college faculty (2017) was obtained from the National Center for Education Statistics [31]. Demographic information for people with earned doctorates was obtained from the NSF 2018 Survey of Earned Doctorates (https://www.nsf.gov/statistics/srvydoctorates/).

## Statistical methods

We used descriptive statistics and data visualization to examine the characteristics of the analytic data set and to examine representation of female authors at the reading, course, and discipline levels. There were three readings with a non-binary first author. Due to small sample size and consistent with other similar research, the research team decided to drop these three observations. Before dropping the data, we examined the three readings by non-binary authors. All three readings were sole authored. Two of the readings, a book and a journal article, were from the same upper-division course in the Women, Gender, and Sexuality Studies department, which is part of the humanities discipline. The third reading was from a graduate-level course in Social Work, which is in the social sciences discipline. Both course instructors were female.

## Results

The final sample included 2435 readings from 129 unique courses in 39 departments from across 6 schools (Table 1). The readings represented the four disciplines sampled: humanities (n = 1130; 46.4%), social sciences (n = 939; 38.6%), STEM (n = 112; 4.6%), and other (n = 254; 10.4%). While the number of readings per discipline varied widely, the number of courses per discipline had a narrower range from 20 for *other* to 44 for *social sciences*, suggesting a much higher number of readings per course in the humanities (mean = 29.7) and social science (mean = 21.3) disciplines compared to the STEM (mean = 4.1) and other disciplines (mean = 12.7).

There were 3961 students enrolled in the 129 unique courses. Some students may be enrolled in more than one course in our sample, so it is likely that the total number of students represented in these courses is somewhat lower than this. Course enrollment ranged from 1 to 346 students with a mean of 30.7 and a median of 15 per class. Class size varied by discipline,

**Table 1. Representation of female authors on 2435 readings from 129 syllabi for 2018–2019 college courses at a large private university.**

|  | All disciplines | Humanities | Social sciences | STEM | Other |
|---|---|---|---|---|---|
| **Total** | | | | | |
| Number of courses | 129 | 38 | 44 | 27 | 20 |
| Number of readings | 2,435 | 1,130 | 939 | 112 | 254 |
| Number of students | 3,961 | 441 | 878 | 1,991 | 651 |
| Number of authors | 4,105 | 1,274 | 2,145 | 239 | 447 |
| Number of female authors | 1,509 | 457 | 954 | 34 | 64 |
| Course characteristics | | | | | |
| Mean number of readings | 19 | 30 | 21 | 4 | 13 |
| Mean number of students | 31 | 12 | 20 | 74 | 33 |
| Percent with female instructors | 41 | 61 | 48 | 22 | 15 |
| Reading characteristics | | | | | |
| Mean percent of female authors | 34 | 36 | 40 | 16 | 12 |
| Median percent of female authors | 0 | 0 | 33 | 0 | 0 |
| Number with female first/sole author | 822 | 404 | 376 | 15 | 27 |
| Percent with female/sole first author | 34 | 36 | 40 | 13 | 11 |
| Readings in classes with female instructors | | | | | |
| Mean percent of female authors | 48 | 49 | 49 | 27 | 17 |
| Median percent of female authors | 50 | 50 | 50 | 0 | 0 |
| Number with female first/sole author | 563 | 299 | 251 | 11 | 2 |
| Percent with female first/sole author | 49 | 50 | 51 | 25 | 14 |
| Readings in classes with male instructors | | | | | |
| Mean percent of female authors | 21 | 21 | 29 | 8 | 12 |
| Median percent of female authors | 0 | 0 | 0 | 0 | 0 |
| Number with female first/sole author | 259 | 105 | 125 | 4 | 25 |
| Percent with female first/sole author | 20 | 20 | 28 | 6 | 10 |

with smaller humanities courses having a narrower range between the smallest class (n = 4 students) and the largest class (n = 31 students); humanities classes had a mean of 11.6 and a median of 11 students per class. Social science courses ranged from 1 to 87 students with a mean of 20 and a median of 16.5 per class. STEM courses had higher enrollment, with class size ranging from 5 to 346 students with a mean of 73.7 and a median of 31 per class. Other discipline classes ranged from 7 students to 118 students, with a mean of 32.5 and a median of 21.5 students per class.

## Total representation of authors by gender

We computed the percentage of authors for each reading who were female. The mean percentage of female authors per reading was 34.1% and the median percentage was 0%. Of the 2435 readings, 1416 had no female authors, while 657 had only female authors. In addition to examining the percentage of female authors per reading, we examined whether each reading had a first author or sole author who was female. We found that 822 of 2435 (33.8%) had a female first or sole author. The remaining 1613 (66.2%) had a male first or sole author.

## Gender of authors by discipline

We examined percentage of authors for each reading who were female within each discipline. In humanities courses, 704 of the 1130 readings had no female authors, while 391 of the 1130 had only female authors. Of the 939 readings in social science courses, 424 had no female

authors, while 239 had only female authors. Of the 112 readings in STEM, 86 had no female authors, while 12 had only female authors. Of the 254 readings in other discipline courses, 202 had no female authors, while 15 had only female authors. While exploring project limitations, we compared syllabi retrieved from the university archive to the syllabi received from faculty through email and found that the percentage of first or sole authors who were female was higher in the emailed syllabi than in the archival syllabi for all four disciplines. Specifically, there were 22.2%, 7.7%, 38.0%, and 12.1% readings with female first or sole authors in archival humanities, other, social sciences, and STEM syllabi respectively and 44.3%, 10.9%, 46.1%, and 18.2% in emailed syllabi for humanities, other, social sciences, and STEM syllabi respectively. We examine this potential source of bias further in the discussion.

Social sciences had the highest mean percentage of female authors per reading (mean = 39.8%) and the highest median percentage of 33.3% female authors per reading. The mean and median for humanities were 36.1% and 0%, for STEM were 15.6% and 0%, and for other were 12.1% and 0%. The only discipline with a non-zero median percentage of female authors per reading was social science. The pattern persisted when we examined percentage of readings with a female first or sole author (i.e., female-led authorship) by discipline (Fig 2), with all four disciplines having a higher percentage of male first/sole authors compared to female first/sole authors and social sciences having the highest percentage of readings with female first/sole authors.

## Gender of authors by discipline and instructor gender

Most courses had a single instructor (n = 121) with 8 courses having 2 or more instructors. Nearly half of courses (n = 53) had a female instructor as the only or lead instructor. Of the 1158 readings in the 53 courses with a female lead instructor, 498 had no female authors, while 459 had only female authors. Of the 1277 readings in the 76 courses with a male lead instructor, 918 had no female authors, while 198 had only female authors.

We examined the mean and median percentage of female authors per reading in courses by the gender of the first instructor listed for the course and course discipline (Fig 3). Female instructors assigned readings with a higher mean percentage of female authors across all disciplines. Courses in the social sciences and humanities with female instructors were the only courses with a median percentage of female authors above zero. Likewise, female instructors in all disciplines assigned a higher percentage of readings with a female first/sole author compared to male instructors (Fig 4).

In light of the potential for more female authorship of book chapters compared to males [4], we examined reading type but found that there were too few book chapters (n = 48; 2%) for a meaningful comparison across disciplines and author gender (see S1 Appendix for more detail on reading type). The small number of book chapters seems consistent with the lower status of chapters as a scholarly product.

## Comparing representation in readings to discipline authorship and workforce

First, we compared the representation of female authors in courses by discipline with the representation of female authors from publications indexed in JSTOR from 1545–2011 reported in Table 1 in West and colleagues [2]. We classified each topic in the table into the four disciplines; where scholarship topics in the table were not an exact match to the topics of courses in our sample, we used the closest matching topic or guidance from the National Science Foundation. We found that 24% of authors on humanities papers were female, 25% of authors on other discipline papers were female, 26% for STEM, and 30% for social sciences.

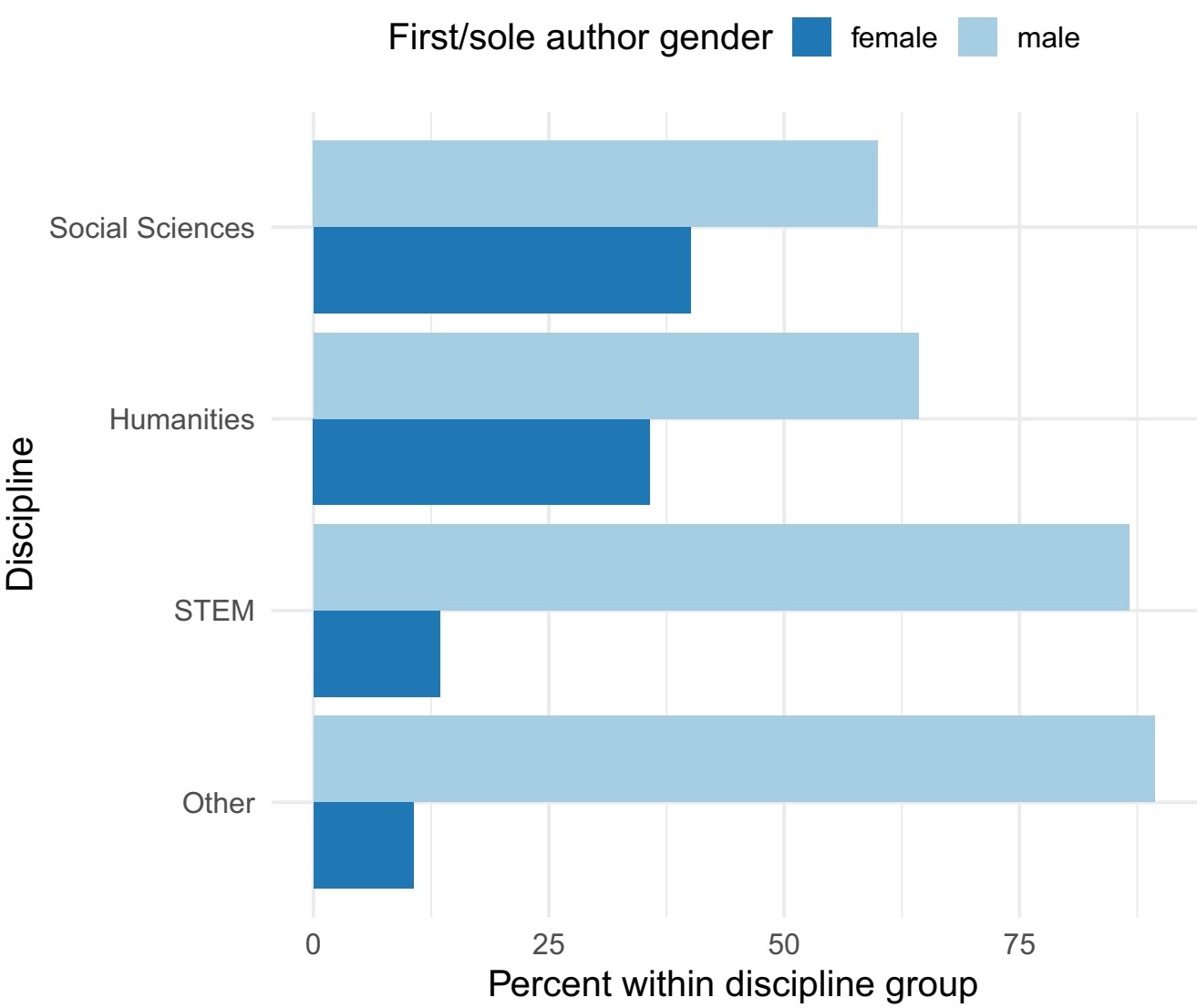

**Fig 2. First/sole author gender by discipline for 2435 readings from syllabi at a large university in 2018.**

Female authors were better represented in the social sciences in our study than in the JSTOR data (44% v. 30%) and in humanities (36% v. 24%), however, women were underrepresented in our data compared to the percentage of articles they have published in STEM (14% v. 26%) and other disciplines (14% v. 25%).

Next, Hardt and colleagues [23] found that male and female faculty in departments with more female faculty assigned more female authors. Although our sample size was too small to test this at the department level, we obtained the number of male and female faculty in each department at WUSTL and computed the percentage of female faculty by discipline. We found that 53.8% of humanities faculty, 52.7% of social sciences faculty, 40.8% of other disciplines faculty, and 25.0% of STEM faculty were female. Women were better represented as faculty in all disciplines than they were in readings in courses in the same discipline.

We also examined the distribution of women in the workforce across the four disciplines to determine if representation of female authors in courses was consistent with representation of women in the workforce across the four major discipline categories. We used data from the

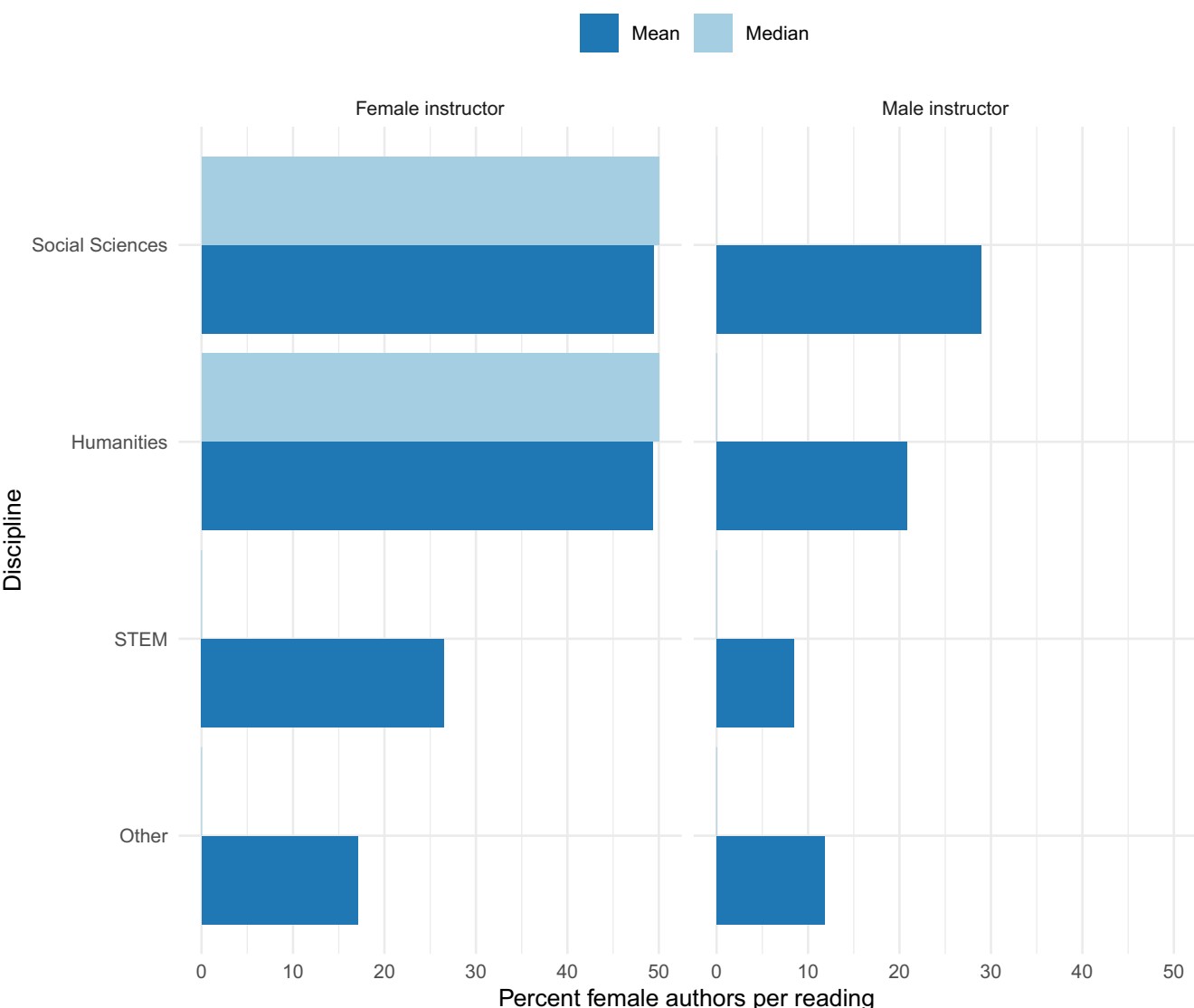

**Fig 3. Mean and median percent female authors per reading by discipline on syllabi from a sample of 129 courses at a large university in 2018 (missing bars indicate median value of 0).**

2017 National Survey of College Graduates. We found that 27.3% of surveyed college graduates employed in STEM fields were female, while 72.7% in STEM fields were male. Female college graduates surveyed were 54.6% of the humanities workforce, 70.8% of the social sciences workforce, and 49.7% of the workforce in other disciplines. Women were better represented in all fields than they were in readings in courses in the same discipline.

In terms of the faculty workforce, as of 2017, 46.3% of faculty at degree-granting institutions in the United States were female [31], with 43.2% of faculty on the tenure track/tenured being female. This is comparable to the proportion of female instructors in our sample (41.1%). We examined the representation of women in the NSF survey of earned doctorates in the US in 2018 to offer some perspective on differences in representation between the representation of female authors in college course readings and the representation of females among recent graduates who might enter the academic workforce. Of the 1,298 who participated in the earned doctorates survey, females earned 46.7% of the doctorates in STEM, 49.4% in social

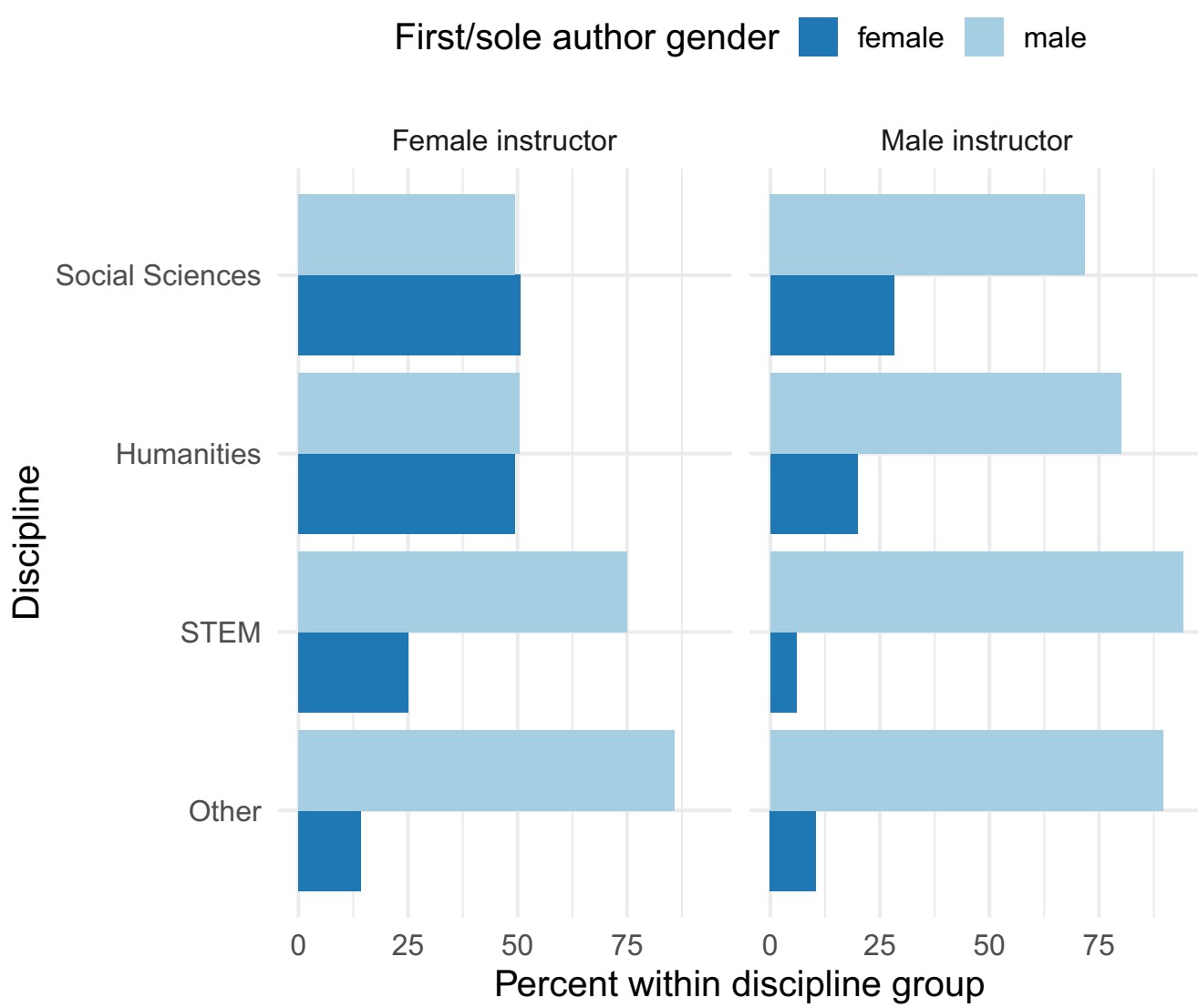

**Fig 4. Percentage of readings with a female first/sole author by instructor gender and discipline on syllabi from a sample of 129 courses at a large university in 2018.**

sciences, 53.2% in humanities, and 54.8% in other fields. As with the college graduates, women were better represented in faculty roles and in all fields of earned doctorates than they were in readings in courses.

## Discussion and conclusions

We examined the authorship of 2,435 readings from the syllabi of 129 courses taught across campus during the 2018–2019 academic year at a large private university. We found evidence of a syllabi gender gap with female authors underrepresented on syllabi as sole and first authors and as members of authorship teams. There were differences in representation by discipline and by gender of the instructor, with female instructors in social sciences and humanities disciplines choosing half of readings with female first or sole authors and having about half of authors per reading who were female. No other combination of gender and discipline had

more than 28% of readings with female first or sole authors; STEM courses taught by male instructors remarkably included just 6% of readings with female first or sole authors.

Our study contributes to the growing evidence base suggesting a sizeable syllabi gender gap [17, 22–26] that varies by instructor gender and discipline. The results also reveal one of the ramifications of women being less likely than men to author peer-reviewed publications [2]. Systemic changes in the peer-review process, such as improving gender equity in reviewers and editorial board membership [5–9], may foster improvements in the number of publications by women scholars which could increase representation in assigned readings. Fewer female authors on syllabi is also a disadvantage to students. This is particularly relevant for disciplines with larger gender gaps such as STEM fields. Assigned readings by female authors can improve exposure to gender-diverse role models and thereby reduce gender stratification and promote greater workforce diversity in these fields [19–21].

Our study has several limitations. First, the classification of gender as only male or female is a limitation of the genderize API that resulted in a likely extreme undercount of non-binary authors. There may have also been some misclassification of male and female gender by the genderize API and by the research team. Additionally, consistent with some other work on gender and syllabi [25], our sample was limited to a single campus and the final sample size was too small to compute meaningful department-level estimates and examine differences among departments within disciplines. However, by collecting syllabi from a single campus, we were able to make comparisons between disciplines. For future scholars wishing to build on this work, we would recommend considering sampling departments as well as disciplines to provide department-level information.

Finally, our data may be biased toward syllabi with more female authors. Including the topic of the study in the email requesting syllabi may have biased responses. While most responses were simply attachments with a neutral message like "here it is," we did receive 15 response emails that reacted positively to the topic and one that reacted negatively. Given this, we believe any bias would be in the direction of overrepresenting syllabi from those who are supportive of gender diversity, which could bias our results toward more representation. We compared the syllabi retrieved from the university archive to the syllabi received from faculty through email and found that the percent of first or sole authors who were female was higher in the emailed syllabi than in the archival syllabi for all four disciplines (see S2 Appendix for table) with 22.2%, 7.7%, 38.0%, and 12.1% female first authors in archival humanities, other, social sciences, and STEM syllabi respectively compared to 44.3%, 10.9%, 46.1%, and 18.2% in emailed syllabi for humanities, other, social sciences, and STEM syllabi respectively. The percentages in the archived syllabi are more in line with West et al. [2] and seem to confirm that the emailed syllabi may overrepresent the percentage of female first or sole authors on syllabi at WUSTL. Despite these limitations, this is the first study that we know of to examine the gender of authors assigned in courses across disciplines at a large university and compare gender representation in college syllabi to gender representation in the workforce.

Given the persistent limited representation of women in STEM and other disciplines, our first recommendation is to improve awareness of the gender syllabi gap among faculty. Numerous emails from faculty sending us their syllabi remarked that they had not thought of the gender of authors when selecting readings and would start paying attention to this after receiving our email request. This suggests that simple awareness may result in some improvement. One faculty member from a humanities department seemed to realize for the first time that all of their authors were female and planned to add work by male authors to improve the diversity of voices students were exposed to in class.

Our second recommendation is to more widely examine representation on syllabi in order to better understand the patterns of representation across higher education. This

recommendation will require overcoming some of the data-related barriers we faced during this project including the underutilization of the central syllabus repository and the immense variation in how faculty format syllabi. Encouraging faculty to use full citations for readings on syllabi and adopting and enforcing university policy to collect syllabi in a central location would be first steps in this process. Development of more sophisticated technological tools for automating the examination of syllabi might be a longer-term and more resource intensive solution.

Finally, to aid faculty in identifying readings by underrepresented groups more easily, we recommend the development of reading collections that faculty can draw from. These collections could ideally be housed in a central location, perhaps on discipline-specific professional association websites or discipline neutral locations like the website for *The Chronicle of Higher Education*.

## Supporting information

**S1 Data. A de-identified version of the syllabus data set.**
(CSV)

**S2 Data. National Science Foundation 2017 survey of college graduates data.**
(CSV)

**S3 Data. NSF 2018 survey of earned doctorates.**
(CSV)

**S4 Data. Washington University in St. Louis number of male and female faculty by discipline.**
(CSV)

**S1 Codebook. A codebook for the de-identified version of the syllabus data set.**
(DOCX)

**S1 File.**
(R)

**S1 Appendix. Text, table, and figure showing the number and percentage of four different reading types by discipline and author gender.**
(DOCX)

**S2 Appendix. Table showing percentage of female first authors on syllabi collected through archival sources compared to syllabi collected by direct email to instructors.**
(DOCX)

## Acknowledgments

The authors would like to thank Lisa Wiland for providing institutional data and Paaige K. Turner, Nancy Cheak-Zamora, and Vetta L. Sanders Thompson for advice on terminology and some of the choices we made in reporting our results. We would also like to acknowledge the reviewers who gave excellent specific suggestions that improved the manuscript.

## Author Contributions

**Conceptualization:** Jenine K. Harris, Merriah A. Croston, Ellen T. Hutti, Amy A. Eyler.

**Data curation:** Jenine K. Harris, Merriah A. Croston, Ellen T. Hutti, Amy A. Eyler.

**Formal analysis:** Jenine K. Harris, Merriah A. Croston, Ellen T. Hutti.

**Methodology:** Jenine K. Harris, Merriah A. Croston, Ellen T. Hutti.

**Project administration:** Jenine K. Harris.

**Software:** Jenine K. Harris, Merriah A. Croston, Ellen T. Hutti.

**Supervision:** Jenine K. Harris, Amy A. Eyler.

**Visualization:** Jenine K. Harris.

**Writing – original draft:** Jenine K. Harris, Merriah A. Croston, Amy A. Eyler.

**Writing – review & editing:** Jenine K. Harris, Merriah A. Croston, Ellen T. Hutti.

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
