## [Decision Letter · Decision Letter 0]

16 Jun 2020

PONE-D-20-13578

Diversify the syllabi: Underrepresentation of female authors in college course readings

PLOS ONE

Dear Dr. Harris,

Thank you for submitting your manuscript to PLOS ONE. After careful consideration, we feel that it has merit but does not fully meet PLOS ONE’s publication criteria as it currently stands. Therefore, we invite you to submit a revised version of the manuscript that addresses the points raised during the review process.

As you can see, the reviewers have arrived at something of a split decision.  Nonetheless, I see a good deal of overlap in the advice they are offering.  Both are pushing you to provide a stronger theoretical framing for the empirical exercise you have conducted.  Reviewer #2 is a bit more explicit about directions in which you might pursue this effort, but I urge you to give consideration to the suggestions of both reviewers.  A revised submission that responds to the these suggestions will be a more valuable contribution to the literature and appropriate for PLOS ONE.

We look forward to receiving your revised manuscript.

Kind regards,

Joshua L Rosenbloom

Academic Editor

PLOS ONE

Journal Requirements:

Additional Editor Comments (if provided):

Reviewers' comments:

Reviewer's Responses to Questions

**Comments to the Author**

1. Is the manuscript technically sound, and do the data support the conclusions?

Reviewer #1: No

Reviewer #2: Yes

2. Has the statistical analysis been performed appropriately and rigorously? 

Reviewer #1: No

Reviewer #2: Yes

3. Have the authors made all data underlying the findings in their manuscript fully available?

Reviewer #1: Yes

Reviewer #2: Yes

4. Is the manuscript presented in an intelligible fashion and written in standard English?

Reviewer #1: Yes

Reviewer #2: Yes

5. Review Comments to the Author

Reviewer #1: PONE-D-20-13578

Diversify the syllabi: Underrepresentation of female authors in college course readings

PLOS ONE

Recommended is theoretical and empirical development for this manuscript.

1) Theoretically, the framework appears to that of “gender stratification.” Addressing the analytical issues within the framework will set a context for the paper and develop an argument.

2) Empirically, the manuscript will be developed by considering the women authors in readings, by levels of courses and types of readings. These, in turn, will link to broader issues (above) of gender stratification.

a) More specifically, address women authors

1) appearing in syllabi in these levels of courses:

-Undergraduate courses: lower-division

-Undergraduate courses: upper-division

-Graduate courses

2) and appearing in text books compared to readings outside of textbooks.

Retain analyses, by field, for 1) and 2)

b) In implications:

1) Assigned readings in upper division, and especially graduate-level courses, have more bearing on modeling for scholarship and workforce diversity (linked to gender stratification).

2) Assigned reading that appear in scholarly journals and books (compared to textbooks) have more bearing on modeling for scholarship and workforce diversity (linked to gender stratification).

3) In addition, for continuing research (not necessarily that for present manuscript):

Particularly revealing would be analysis of readings, by gender of author, that appear in reading lists for doctoral exams. These are influential for career-bound students.

Reviewer #2: A note: I’m signing this review so that I can discuss my own research more directly, without skirting around my own identity. My work that I’ll discuss below is found in these two citations (which aren’t cited in this article, incidentally):

• Hardt, Heidi, Amy Erica Smith, Hannah June Kim, and Philippe Meister. 2019. “The Gender Readings Gap in Political Science Graduate Training.” The Journal of Politics 81 (4): 1528–32. https://doi.org/10.1086/704784.

• Smith, Amy Erica, Heidi Hardt, Philippe Meister, and Hannah June Kim. 2020. “Gender, Race, Age, and National Origin Predict Whether Faculty Assign Female-Authored Readings in Graduate Syllabi.” PS: Political Science & Politics 53 (1): 100–106. https://doi.org/10.1017/S1049096519001239.

This is the first study of which I am aware to assess the representation of female-authored readings in courses across a wide range of disciplines. As such, it builds upon and makes an important contribution to an increasingly solid body of evidence that syllabi insufficiently assign research by women. The paper should be published in PLOS ONE, with some revisions that should not be terribly burdensome.

The real novelty of this paper is that it draws from multiple disciplines. That said, it is also limited in what it says about those various disciplines. If possible, I would urge the authors to consider contextual effects: what makes work by women more likely to be assigned in social sciences than in other fields? In Hardt et al. (2019) (see above citation), we try to take advantage of variation across both subfields of political science and departments to assess what we call the “supply side” and “demand side” correlates of assigning work by women. First, we find that subfields of political science that have more women scholars also have a higher rate of assigning work by women—but there’s a lot that can’t be explained by women’s presence alone. Second, we find that in departments with more female faculty, both male and female instructors are more likely to assign work by women; we argue that this is due to changing networks and norms. It seems like it would be possible to test both of these hypotheses—or likely others we didn’t think of—in your data. It’s not absolutely necessary for publishing this current piece, but these are things to think about.

In addition, you’ll find in the Smith et al. (2020) article that we branch out beyond looking only at instructor gender to identify other instructor characteristics that shape “demand” for female-authored scholarship. Most intriguing to me is that we find that race and gender intersect; men of color are indistinguishable from female scholars in assigning rates of female scholarship. Again, I don’t think you have to address all these potential hypotheses now in PLOS ONE, but I’d urge you to push your data farther.

A second issue where I DO think some more work is needed before publication is on the baseline. That is, how do we know whether work is underassigned? You treat percentage of women among the workforce as one potential baseline, but skeptics are going to point out instructors can’t assign things that haven’t been written, and that the average worker in the humanities workforce isn’t writing college textbooks. Likewise, earned doctorates aren’t quite an appropriate baseline for similar reasons, though that baseline is getting closer. The percentage of faculty who are female is even better, but skeptics will still be dissatisfied because we already know that women publish at somewhat lower rates than do men. So, the best baseline would be a measure of the proportion of potentially assignable publications that are female-authored. Obviously it may be very hard to get those data for all disciplines – but, actually, look at the West et al. piece you cite. It may have the numbers you want.

In the conclusion, I’d like to have the authors discuss limitations of the sampling strategy. First, I’m concerned that something about the recruitment email may have induced bias in the response rate, given that some faculty responded by talking about gender. I am afraid that you may have inadvertently turned off some faculty who are not sympathetic to the goals of the project. Second, limiting the study to Wash U has some strengths, in that Wash U provided a convenient data source, and also that the choice made a comparison across disciplines feasible. Still, I’d like to have the authors reflect on and report back on whether there’s anything about Wash U that might affect the data. To what extent can we expect Wash U to be representative of the median university in the US? In that regard, see Phull et al. (2018), which I list below; their study is limited just to the LSE, and provides a little bit of precedent for studying a single institution.

Finally, I’m concerned about the way nonbinary status is treated. This coding strategy is likely to dramatically underrepresent transgender and nonbinary people, making it very hard to draw any conclusions about nonbinary authors. The strategy primarily relies on machine coding of given names based on whether they are traditionally female or traditionally male, and there is no such thing as a traditionally nonbinary name. Hence, nonbinary is only a POSSIBLE category for a small percentage of respondents who were hand-coded. Moreover, we have no idea of whether work by nonbinary scholars is over- or under-assigned, since we don’t have baseline data on the proportion of nonbinary scholars in relevant populations. Because of all of these issues, in my work and previous work before us, we’ve excluded the category of nonbinary. That’s really not a good solution either, but the strategy in this article gives the appearance but not reality of accounting for nonbinary status. I do feel sure that lumping nonbinary scholars in with women is likely to offend someone. There’s no good solution here, but I would like the authors to think this through and make some changes.

Small things:

• The “percent of female authors” number is confusing without a control for number of authors. It may be clearer simply to focus on female first/only authors.

• The “male” bars in the figures can probably be deleted for the sake of simplicity, since “male” and “female” are effectively inverses of each other, as there are only three nonbinary readings.

In addition to my own work, here are two other prior studies in specific subfields of political science that deal with precisely this topic:

• Diament, Sean M., Adam J. Howat, and Matthew J. Lacombe. 2018. “Gender Representation in the American Politics Canon: An Analysis of Core Graduate Syllabi.” PS: Political Science & Politics 51 (3): 635–40. https://doi.org/10.1017/S1049096518000392.

• Phull, Kiran, Gokhan Ciflikli, and Gustav Meibauer. 2018. “Gender and Bias in the International Relations Curriculum: Insights from Reading Lists.” European Journal of International Relations, August. https://doi.org/10.1177/1354066118791690.

6. PLOS authors have the option to publish the peer review history of their article (what does this mean?). If published, this will include your full peer review and any attached files.

Reviewer #1: No

Reviewer #2: Yes: Amy Erica Smith

---

## [Author Response · Author response to Decision Letter 0]

17 Jul 2020

PLEASE NOTE WE ATTACHED A VERSION OF THIS WITH THE RESUBMISSION LETTER; IT HAS SEVERAL GRAPHS AND TABLES THAT CANNOT BE SEEN IN THIS TEXT-ONLY FIELD.

Response to Reviewers: PONE-D-20-13578

Reviewer #1

Recommended is theoretical and empirical development for this manuscript.

1) Theoretically, the framework appears to that of “gender stratification.” Addressing the analytical issues within the framework will set a context for the paper and develop an argument.

Thank you for this suggestion, we agree that the syllabi gap contributes to gender stratification in academia and have added the following statement with citations of papers describing gender stratification to the introduction (page 4, lines 55-72):

These patterns of in scholarly productivity and visibility result in differences in salary, promotion, and tenure in academia [3], contributing to the gender stratification [16] of academic career success [3,4,17,18] (https://www.chronicle.com/article/Divvying-Up-the-Raise-Pool/45750/).

We also added a statement about gender stratification to the conclusion.

2) Empirically, the manuscript will be developed by considering the women authors in readings, by levels of courses and types of readings. These, in turn, will link to broader issues (above) of gender stratification.

a) More specifically, address women authors

1) appearing in syllabi in these levels of courses:

-Undergraduate courses: lower-division

-Undergraduate courses: upper-division

-Graduate courses

We added a new variable for each course categorizing it as lower division or upper division + graduate courses (since some courses accept both types of students). We compared the percentage of readings on syllabi for lower division and upper/grad and found that lower division courses have a higher percentage of female first or sole authored readings in three of the four disciplines, while the other discipline courses did not have lower division courses in the sample (Fig). 

We identified a paper (suggested by Reviewer 2) that supported the idea that syllabi have a role in socializing doctoral students (Smith et. al., 2020), but it did not cite additional evidence nor compare the role of syllabi in socializing future faculty with syllabi throughout the educational process widening the gender gap in the non-academic workforce. 

After discussions in the authorship team we are not convinced that representation of women as authors on upper division and graduate school syllabi are more linked to representation in the workforce than representation on lower division syllabi (although it may be linked to representation in academia). We added a sentence about doctoral students and cited the Smith paper (page 5, lines 79-81): 

In addition to encouraging students earlier in the educational process, course readings on syllabi in doctoral courses may be socializing the next generation of faculty (Smith et. al., 2020). 

We appreciate this suggestion and will keep thinking about it as we move forward with this work.

2) and appearing in text books compared to readings outside of textbooks.

The type of reading in this study included a lot of different sorts of documents including textbooks, popular books, book chapters, articles from magazines, articles from newspapers, blogs, journal articles, poems, and more. Although this was not among our priorities in collecting the data, we were able to use our notes to classify each reading as a book, book chapter, journal article, or other to explore these data in light of the reviewer suggestion. However, classifying the book reading type into textbook and other types of books is beyond the scope of the current project data. 

The research team was unaware that although women publish fewer journal articles, women may submit and/or publish an equal number or more book chapters compared to men (Mayer & Rathmann, 2018; Djupe, Smith, & Sokhey, 2019). Given the lower status of chapter publishing compared to journal article publishing, this may contribute to limitations on success for promotion and tenure.

In our data, we found consistency with Mayer & Rathman and Djupe et al. with social sciences and other discipline having the most female authorship of book chapters, followed by journal articles, and then books. Humanities was similar to social sciences and other with the exception of the authorship of other reading types, which was larger. However, given the extremely small numbers of book chapters assigned across the syllabi (just under 2% of readings; n = 48), we decided not to include this in the main text of the paper, but instead added the following text, table, and figure in Appendix 1.

Finally, given the findings of Mayer and Rathmann (2018) and Djupe et al. (2019) we examined type of reading by discipline and author gender. We found that, in social sciences and humanities, there was a higher percentage of female first/sole authors than male for book chapters and in the other disciplines, book chapters had the highest proportion of female authors of any of the types of readings. No book chapters were included on STEM syllabi.

Table. Number of readings of different types by discipline.

category reading_type_clean count

 <chr> <chr> <int>

 1 Humanities book 469

 2 Humanities book chapter 6

 3 Humanities journal article 199

 4 Humanities other 441

 5 Other book 84

 6 Other book chapter 4

 7 Other journal article 101

 8 Other other 63

 9 Social Sciences book 261

10 Social Sciences book chapter 38

11 Social Sciences journal article 503

12 Social Sciences other 126

13 STEM book 77

14 STEM journal article 20

15 STEM other 11

Fig. Percentage of each type of reading with a female first or sole author within each discipline. 

We also added this text to the main text (page 16; line 528):

In light of the potential for more female authorship of book chapters compared to males [4], we examined reading type but found that there were too few book chapters (n = 48; 2%) for a meaningful comparison across disciplines and author gender (see Appendix 1 for more detail on reading type). The small number of book chapters seems consistent with the lower status of chapters as a scholarly product.

Retain analyses, by field, for 1) and 2)

b) In implications:

1) Assigned readings in upper division, and especially graduate-level courses, have more bearing on modeling for scholarship and workforce diversity (linked to gender stratification).

See response above. 

2) Assigned reading that appear in scholarly journals and books (compared to textbooks) have more bearing on modeling for scholarship and workforce diversity (linked to gender stratification).

See response above. 

3) In addition, for continuing research (not necessarily that for present manuscript):

Particularly revealing would be analysis of readings, by gender of author, that appear in reading lists for doctoral exams. These are influential for career-bound students.

Thank you for the suggestion. It was beyond the scope of our current work and we have added it to our growing list of research questions for our ongoing efforts in this area.

Reviewer #2 

A note: I’m signing this review so that I can discuss my own research more directly, without skirting around my own identity. My work that I’ll discuss below is found in these two citations (which aren’t cited in this article, incidentally):

• Hardt, Heidi, Amy Erica Smith, Hannah June Kim, and Philippe Meister. 2019. “The Gender Readings Gap in Political Science Graduate Training.” The Journal of Politics 81 (4): 1528–32. https://doi.org/10.1086/704784.

• Smith, Amy Erica, Heidi Hardt, Philippe Meister, and Hannah June Kim. 2020. “Gender, Race, Age, and National Origin Predict Whether Faculty Assign Female-Authored Readings in Graduate Syllabi.” PS: Political Science & Politics 53 (1): 100–106. https://doi.org/10.1017/S1049096519001239.

Thank you for the suggestions, we have added these citations, among others, to our introduction and conclusion.

This is the first study of which I am aware to assess the representation of female-authored readings in courses across a wide range of disciplines. As such, it builds upon and makes an important contribution to an increasingly solid body of evidence that syllabi insufficiently assign research by women. The paper should be published in PLOS ONE, with some revisions that should not be terribly burdensome.

The real novelty of this paper is that it draws from multiple disciplines. That said, it is also limited in what it says about those various disciplines. If possible, I would urge the authors to consider contextual effects: what makes work by women more likely to be assigned in social sciences than in other fields? In Hardt et al. (2019) (see above citation), we try to take advantage of variation across both subfields of political science and departments to assess what we call the “supply side” and “demand side” correlates of assigning work by women. First, we find that subfields of political science that have more women scholars also have a higher rate of assigning work by women—but there’s a lot that can’t be explained by women’s presence alone. Second, we find that in departments with more female faculty, both male and female instructors are more likely to assign work by women; we argue that this is due to changing networks and norms. It seems like it would be possible to test both of these hypotheses—or likely others we didn’t think of—in your data. It’s not absolutely necessary for publishing this current piece, but these are things to think about.

We were able to obtain the number of male and female faculty each department, but the micronumerosity problem persisted. We did compute the percentage of faculty who were female by discipline and compared it to the syllabi authorship data. We added this to the section on representation, with a citation to the Hardt et al. 2019 paper (page 17, lines 546-552):

Next, Hardt and colleagues [23] found that male and female faculty in departments with more female faculty assigned more female authors. Although our sample size was too small to test this at the department level, we obtained the number of male and female faculty in each department at WUSTL and computed the percentage of female faculty by discipline. We found that 53.8% of humanities faculty, 52.7% of social sciences faculty, 40.8% of other disciplines faculty, and 25.0% of STEM faculty were female. Women were better represented as faculty in all disciplines than they were in readings in courses in the same discipline.

In addition, you’ll find in the Smith et al. (2020) article that we branch out beyond looking only at instructor gender to identify other instructor characteristics that shape “demand” for female-authored scholarship. Most intriguing to me is that we find that race and gender intersect; men of color are indistinguishable from female scholars in assigning rates of female scholarship. Again, I don’t think you have to address all these potential hypotheses now in PLOS ONE, but I’d urge you to push your data farther.

We were not able to obtain the race of the specific instructors included in the sample and the closest we could get from archival data at the department level was percent white, but this point is well-taken and was something we thought about examining at the onset of the project. It remains on our list of topics to work on and our team would welcome collaboration opportunities if you have any interest. 

A second issue where I DO think some more work is needed before publication is on the baseline. That is, how do we know whether work is underassigned? You treat percentage of women among the workforce as one potential baseline, but skeptics are going to point out instructors can’t assign things that haven’t been written, and that the average worker in the humanities workforce isn’t writing college textbooks. Likewise, earned doctorates aren’t quite an appropriate baseline for similar reasons, though that baseline is getting closer. The percentage of faculty who are female is even better, but skeptics will still be dissatisfied because we already know that women publish at somewhat lower rates than do men. So, the best baseline would be a measure of the proportion of potentially assignable publications that are female-authored. Obviously it may be very hard to get those data for all disciplines – but, actually, look at the West et al. piece you cite. It may have the numbers you want.

Thank you for this suggestion. We added a paragraph comparing female authorship from Table 1 in the West article to female authorship in our data. The West et. al. data were not perfectly aligned with the disciplines in our study and were nearly all research articles, while the syllabi we reviewed included quite a few books and a number of other types of documents like poetry or newspaper articles. Nevertheless, the paragraph provides another, and perhaps more relevant, source of comparison between what is being taught and what is actually going on in the field. We also added three rows to the table showing the number of total authors by discipline and the number and percentage of female authors by discipline. Here is the new paragraph (page 16; lines 533-542):

First, we compared the representation of female authors in courses by discipline with the representation of female authors from publications indexed in JSTOR from 1545--2011 reported in Table 1 in West and friends (2013). We classified each topic in the table into the four disciplines; where scholarship topics in the table were not an exact match to the topics of courses in our sample, we used the closest matching topic or guidance from the National Science Foundation. We found that 23.6% of authors on humanities papers were female, 25.0% of authors on other discipline papers were female, 25.6% for STEM, and 29.8% for social sciences. Women authors were better represented in the social sciences in our study than in the JSTOR data (45% v. 30%) and in humanities (36% v. 24%), however, women were underrepresented in our data compared to the percentage of articles they have published in STEM (14% v. 26%) and other disciplines (15% v. 25%).

In the conclusion, I’d like to have the authors discuss limitations of the sampling strategy. First, I’m concerned that something about the recruitment email may have induced bias in the response rate, given that some faculty responded by talking about gender. I am afraid that you may have inadvertently turned off some faculty who are not sympathetic to the goals of the project. Second, limiting the study to Wash U has some strengths, in that Wash U provided a convenient data source, and also that the choice made a comparison across disciplines feasible. Still, I’d like to have the authors reflect on and report back on whether there’s anything about Wash U that might affect the data. To what extent can we expect Wash U to be representative of the median university in the US? In that regard, see Phull et al. (2018), which I list below; their study is limited just to the LSE, and provides a little bit of precedent for studying a single institution. 

We added context about the University as compared to other universities and added a link to the Phull et al paper in the limitations section. We also added a statement about the strength of being able to compare disciplines. Here is the added text describing the university (page 7, lines 122-135):

WUSTL is a large private secular university situated in St. Louis, Missouri, United States (US). The 2020 total enrollment was 13,654, which includes 2,717 international students. The 2018-2019 WUSTL undergraduate student body was 53% female and 61% US White with 15% of students qualifying for the Federal Pell Grant Program that supports low-income students. In 2020, WUSTL was ranked the 19th best overall of national universities in the US, 31st in best global universities, and 61st best in undergraduate education in the US. WUSTL is highly selective, with 15% of applicants accepted in 2018 and 88% on-time graduation. The student body has more ethnic diversity than most national universities in the US and is among the most selective, but has a very little economic diversity compared to other US universities. The national and global reputation of WUSTL suggest a high-quality institution that is a model for other Universities. The majority of WUSTL programs are housed on one of two main campuses, the Danforth campus and the Medical campus.

We also conducted a sensitivity analysis to compare the percentage of female sole/first authors across the disciplines for those syllabi we retrieved from archival sources to those received in response to our email request. We found that the percentage of female authors on syllabi received in response to our email was notably higher across all the disciplines. We added a description of this sensitivity analysis to our limitations and included the table in an appendix (pages 19-20; lines 608-622):

Finally, our data may be biased toward syllabi with more female authors. Including the topic of the study in the email requesting syllabi may have biased responses. While most responses were simply attachments with a neutral message like “here it is,” we did receive 15 response emails that reacted positively to the topic and one that reacted negatively. Given this, we believe any bias would be in the direction of overrepresenting syllabi from those who are supportive of gender diversity, which could bias our results toward more representation. We compared the syllabi retrieved from the university archive to the syllabi received from faculty through email and found that the percent of first or sole authors who were female was higher in the emailed syllabi than in the archival syllabi for all four disciplines (see Appendix 2 for table) with 22.2%, 7.7%, 38.0%, and 12.1% female first authors in archival humanities, other, social sciences, and STEM syllabi respectively compared to 44.3%, 10.9%, 46.1%, and 18.2% in emailed syllabi for humanities, other, social sciences, and STEM syllabi respectively. This seems to confirm that the emailed syllabi may overrepresent the percentage of female first or sole authors on syllabi at WUSTL. Despite these limitations, this is the first study that we know of to examine the gender of authors assigned in courses across disciplines at a large university and compare gender representation in college syllabi to gender representation in the workforce.

Finally, I’m concerned about the way nonbinary status is treated. This coding strategy is likely to dramatically underrepresent transgender and nonbinary people, making it very hard to draw any conclusions about nonbinary authors. The strategy primarily relies on machine coding of given names based on whether they are traditionally female or traditionally male, and there is no such thing as a traditionally nonbinary name. Hence, nonbinary is only a POSSIBLE category for a small percentage of respondents who were hand-coded. Moreover, we have no idea of whether work by nonbinary scholars is over- or under-assigned, since we don’t have baseline data on the proportion of nonbinary scholars in relevant populations. Because of all of these issues, in my work and previous work before us, we’ve excluded the category of nonbinary. That’s really not a good solution either, but the strategy in this article gives the appearance but not reality of accounting for nonbinary status. I do feel sure that lumping nonbinary scholars in with women is likely to offend someone. There’s no good solution here, but I would like the authors to think this through and make some changes.

Our team discussed this again and discussed with faculty members who are experts in diversity, equity, and inclusion. Based on those discussions, we are changing our treatment of the 3 observations and omitting them from the main analyses, citing your work in support of this strategy. We also added a brief description of the 3 readings in question, so as not to completely omit the non-binary authors from this work (pages 10-11; lines 209-225):

There were three readings with a non-binary first author, due to small sample size and consistent with other similar research, the research team decided to drop these three observations. Before dropping the data, we examined the three readings by non-binary authors. All three readings were sole authored. Two of the readings, a book and a journal article, were from the same upper-division course in the Women, Gender, and Sexuality Studies department, which is part of the humanities discipline. The third reading was from a graduate-level course in Social Work, which is in the social sciences discipline. Both course instructors were female.

Small things:

• The “percent of female authors” number is confusing without a control for number of authors. It may be clearer simply to focus on female first/only authors.

We would prefer to leave these results in the main body of the paper but are open to removing them or moving to an Appendix if the editor and reviewer both feel strongly about this. We feel that the percent of female authors provides additional information about the extent to which students are exposed to women authors at all.

• The “male” bars in the figures can probably be deleted for the sake of simplicity, since “male” and “female” are effectively inverses of each other, as there are only three nonbinary readings.

We understand this concern and agree that the information is redundant, but when we reviewed graphs with only female bars, they were visually less impactful in demonstrating the differences between male and female authorship. We did not make any changes to the manuscript based on this suggestion.

In addition to my own work, here are two other prior studies in specific subfields of political science that deal with precisely this topic:

• Diament, Sean M., Adam J. Howat, and Matthew J. Lacombe. 2018. “Gender Representation in the American Politics Canon: An Analysis of Core Graduate Syllabi.” PS: Political Science & Politics 51 (3): 635–40. https://doi.org/10.1017/S1049096518000392.

• Phull, Kiran, Gokhan Ciflikli, and Gustav Meibauer. 2018. “Gender and Bias in the International Relations Curriculum: Insights from Reading Lists.” European Journal of International Relations, August. https://doi.org/10.1177/1354066118791690.

We added these references and several other related papers to the introduction and conclusion as appropriate. Thank you for the suggestion.

---

## [Decision Letter · Decision Letter 1]

28 Aug 2020

Diversify the syllabi: Underrepresentation of female authors in college course readings

PONE-D-20-13578R1

Dear Dr. Harris,

We’re pleased to inform you that your manuscript has been judged scientifically suitable for publication and will be formally accepted for publication once it meets all outstanding technical requirements.

As you will see, the reviewers remain split in their assessment of your submission; but I believe you have passed the bar for publication in PLOS ONE.  Reviewer #2 makes several suggestions that might help to improve the exposition of your article or make it more impactful.  I encourage you to consider incorporating these into the final manuscript you submit.

Kind regards,

Joshua L Rosenbloom

Academic Editor

PLOS ONE

Additional Editor Comments (optional):

Reviewers' comments:

Reviewer's Responses to Questions

**Comments to the Author**

1. If the authors have adequately addressed your comments raised in a previous round of review and you feel that this manuscript is now acceptable for publication, you may indicate that here to bypass the “Comments to the Author” section, enter your conflict of interest statement in the “Confidential to Editor” section, and submit your "Accept" recommendation.

Reviewer #1: (No Response)

Reviewer #2: (No Response)

2. Is the manuscript technically sound, and do the data support the conclusions?

Reviewer #1: Partly

Reviewer #2: Yes

3. Has the statistical analysis been performed appropriately and rigorously? 

Reviewer #1: No

Reviewer #2: Yes

4. Have the authors made all data underlying the findings in their manuscript fully available?

Reviewer #1: Yes

Reviewer #2: Yes

5. Is the manuscript presented in an intelligible fashion and written in standard English?

Reviewer #1: Yes

Reviewer #2: Yes

6. Review Comments to the Author

Reviewer #1: PONE-D-20-13578R1

Diversify the syllabi: Underrepresentation of female authors in college course readings

PLOS ONE

The paper and topic have potential. For publication as an article, the potential needs to be realized.

Missing are key elements:

1) A framework for analysis, results, and conclusions: Sentences added about “gender stratification” do not constitute a framework. Absent are a conceptual and theoretical framework of fundamental gender stratification in academia that is structural, and goes beyond issues of “modeling,” “mentoring,” of “awareness,” related to gender. Consider issues of “gender and hierarchy” in academia.

2) Theoretical and empirical connection: The prior review emphasized the need to connect, theoretically and empirically; and the way that the relationship between gender of authors and levels of courses is a key means toward this. As is, the issue of levels (and thus, hierarchy or stratification) are absent from the Abstract, the Introduction, and the fundamental Table 1 of descriptive features.

3) Distinction between causation and absence of causation: The paper refers to course readings as “resulting in a gender gap” in higher education. Based on the data in the paper, the readings reflect a gender gap (and more fundamentally, a hierarchy) – and may reinforce this. No evidence appears here that a gap/hierarchy is a result of readings.

Reviewer #2: I’m largely satisfied with these changes, which have done a good job of addressing my original concerns. I have just scattered small comments remaining:

• In your initial discussion of nonbinary authors and the genderize procedure, I think it would be helpful to point out briefly that the genderize package has no way to deal with nonbinary authors.

• In Table 1, you have lines for both “Percent of female authors” and “Mean percent of female authors.” I think this is going to confuse people, especially because the numbers are close but not identical. I get how the denominators are different, and why they’re calculated differently, but I think that this will be a bit “in the weeds” for most readers who haven’t dealt with this kind of data. Can you pick one?

• You misspelled “sophisticated” in the penultimate paragraph.

• Given the dramatic difference between the gender representation you found in the emailed versus archival syllabi, I would suggest highlighting the different numbers earlier, and I’m inclined to trust the archival numbers more. It’s also worth noting that the % female authors from the archival syllabi are more in line with the West et al. numbers.

7. PLOS authors have the option to publish the peer review history of their article (what does this mean?). If published, this will include your full peer review and any attached files.

Reviewer #1: No

Reviewer #2: **Yes: **Amy Erica Smith

---

## [Editor Report · Acceptance letter]

24 Sep 2020

PONE-D-20-13578R1 

Diversify the syllabi: Underrepresentation of female authors in college course readings 

Dear Dr. Harris:

I'm pleased to inform you that your manuscript has been deemed suitable for publication in PLOS ONE. Congratulations! Your manuscript is now with our production department. 

Kind regards, 

on behalf of

Dr. Joshua L Rosenbloom 

Academic Editor

PLOS ONE